# Transgene Mapping in Animals: What to Choose?

**DOI:** 10.3390/ijms26104705

**Published:** 2025-05-14

**Authors:** Alexander Smirnov, Maksim Makarenko, Anastasia Yunusova

**Affiliations:** 1Institute of Cytology and Genetics, Siberian Branch of Russian Academy of Sciences (SB RAS), Novosibirsk 630090, Russia; 2Department of Genetics and Life Sciences, Sirius University of Science and Technology, Sirius Federal Territory, Sochi 354340, Russia

**Keywords:** transgene, TAIL-PCR, long-read sequencing, genome walking, pronuclear microinjection

## Abstract

The phenomenal progress in biotechnology and genomics is both inspiring and overwhelming—a classic curse of choice, particularly when it comes to selecting methods for mapping transgene DNA integration sites. Transgene localization remains a crucial task for the validation of transgenic mouse or other animal models generated by pronuclear microinjection. Due to the inherently random nature of DNA integration, reliable characterization of the insertion site is essential. Over the years, a vast number of mapping methods have been developed, and new approaches continue to emerge, making the choice of the most suitable technique increasingly complex. Factors such as cost, required reagents, and the nature of the generated data require careful consideration. In this review, we provide a structured overview of current transgene mapping techniques, which we have broadly classified into three categories: classic PCR-based methods (such as inverse PCR and TAIL-PCR), next-generation sequencing with target enrichment, and long-read sequencing platforms (PacBio and Oxford Nanopore). To aid in decision-making, we include a comparative table summarizing approximate costs for the methods. While each approach has its own advantages and limitations, we highlight our top four recommended methods, which we believe offer the best balance of cost-effectiveness, reliability, and simplicity for identifying transgene integration sites.

## 1. Introduction

Transgenic animals are the backbone of modern biology. It is nothing short of a scientific marvel that foreign DNA can integrate into a genome without direct assistance. However, in many cases, transgene insertion remains a black box—unless we can precisely determine the integration site. While early transgene mapping methods were often laborious and technically challenging, the excitement of discovering an integration site, especially when something was unexpectedly misplaced, was undeniable. One integration might have landed within a coding gene, another—next to a non-coding RNA that had only recently been annotated, sometimes the host gene would interact with transgene and form a hybrid transcript. With the explosion of biotechnology and genomics, a vast array of transgene mapping methods has emerged. The field has progressed tremendously. Between 1990 and 2010, before affordable whole-genome sequencing (WGS) became widely available, scientists had to get creative in their quest to locate transgene insertion sites. Numerous PCR-based methods were invented, falling under the umbrella of “genome walking” [1,2]. In an outdated but impressive 2011 review, Leoni et al. catalogued 53 different genome walking methods [3]. The number of such methods likely exceeds a few hundred by now. Modern transgene mapping involves long-read sequencing with target enrichment and multi-omics approaches [1]. Today, integration sites, local chromatin states, and expression levels can all be analyzed in parallel with unprecedented precision and throughput [4]. Although most reviews adopt a historical, archivist perspective, we will deviate from this approach and instead focus on the practical appeal of mapping methods. Despite the wealth of available catalogs, including a recent comprehensive overview [1], researchers who simply want to sequence their transgenic mouse (or other creature) may find themselves overwhelmed by the sheer variety of available techniques.

The choice of mapping method is not just a matter of financial constraints, but also depends on the type of data one seeks to obtain, including read length, on-target (transgene) coverage, and discovery of accompanying genome rearrangements. So, what is the best approach? Is it expensive to just apply WGS to your mice and what coverage would be enough? Would targeted locus amplification (TLA) be optimal to resolve multicopy concatemers? Should one save costs and rely on thermal asymmetric interlaced PCR (TAIL-PCR), leaving the results to sheer luck (quite literally)? We aim to share our experience and recommendations, focusing on sequencing-based approaches for random transgene insertions in animals and, to some extent, in cultured cells.

### 1.1. Features of Random Transgenic Insertion in Animals

In this review, we focus on transgenic animals in which random integration of DNA is typically achieved via pronuclear injection. A natural question arises: if genotyping can be easily performed using qPCR to distinguish between heterozygotes and homozygotes, why bother identifying the exact integration locus at all (Figure 1)?

From a practical perspective, knowing the integration site can prevent downstream complications, especially in cases involving multiple insertions. A large-scale analysis of F0 mouse founder lines showed that approximately 20% had more than one integration site [5]. Multiple transgene loci may lead to unexpected segregation patterns, complicating both genotyping and phenotype interpretation.

Transgene integration is often influenced by position effect variegation (PEV)—the insertion site can significantly affect transgene expression levels. This is well illustrated in Chinese hamster ovary (CHO) cells, widely used in industrial protein production. Studies have shown that transgene insertions often occur in transcriptionally active regions, which are also prone to structural instability, including rearrangements over time [6,7,8]. As reviewed by Cabrera et al., integration into such regions may enhance expression but can also interfere with endogenous gene regulation [9]. Transgenes may be influenced by regulatory sequences located at considerable distances [10], emphasizing the importance of identifying their integration sites.

Furthermore, studies in mice have shown that nearly half of random integrations could potentially disrupt host gene function, either by inserting into introns or causing deletions of coding exons—for example, 45% (17/38) in report of Yan et al. [11], and 53% (21/40) in another work [12]. A recent study of the widely used Ucp1-Cre mouse line—which exhibits lethality in homozygous animals—revealed that the integration of a BAC transgene resulted in a large deletion and inversion affecting four genes, with potential additional effects on seven neighboring genes [13]. Notably, the presence of an active Ucp1 gene copy, which should not exist in the experimental model, influenced fat tissue homeostasis. Such cases are frequent, and genomic sequencing of established mouse strains often resembles archaeological investigation. According to the Mouse Genome Database, only 5% of over 8000 documented mouse transgenic lines have had their integration sites mapped [12].

Another unanticipated feature of random integration is the cointegration of unrelated DNA fragments. Initially considered rare, such events are now frequently observed thanks to deep genome sequencing in both cell lines and animals [14,15,16]. New quantitative methods analyzing CRISPR/Cas9-induced DNA breaks have shown that DNA is often incorporated at double-stranded break (DSB) ends—at frequencies of 0.1–1% per DSB [17,18]. This includes not only cotransfected DNA (which is expected to be abundant) but also genomic segments, repetitive elements, and regulatory sequences. For example, Geng et al. reported a striking “insertional bingo” event, discovering a ~200 bp fragment of *E. coli* DNA, a ~6 kb Cas9 plasmid backbone, and a local genomic duplication at the Cas9 target site [16].

Following pronuclear microinjection, the DNA repair machinery recognizes linear transgene ends and attempts to resolve DSBs by ligating whatever DNA is available [19]. Most commonly, transgene fragments are joined into concatemers, but integrations can also include plasmid backbones, bacterial genomic DNA, or even telomeric repeats (see recent review [20]). In the well-known hornless cattle case, a 200 bp “Celtic” allele was introduced using transcription activator-like effector nucleases (TALENs), but a plasmid backbone fragment was later discovered during U.S. Food and Drug Administration (FDA) re-evaluation [21]. This contamination could have been identified early using plasmid-specific primers—a practice that should become standard in long-term projects. Another illustrative case is the mouse line described by Chiang et al., in which the transgene was fragmented and inserted into host genome with a 168 bp segment of *Corynebacterium* DNA [22]. This sequence likely originated from the lab environment during DNA preparation. Cointegrations of *E. coli* fragments are very common as well [12,15]. Curiously, Hussmann et al. even identified a 165 bp bovine DNA fragment integrated into a CRISPR/Cas9 reporter in human cells—presumably captured from fetal bovine serum in the culture medium [23]. These findings highlight that the nucleus is a crowded environment, and the risk of foreign DNA integration at DSBs is non-negligible and should be carefully considered during mapping. These risks can potentially be minimized by treating plasmid preps with exonucleases to remove bacterial contaminants and carefully performing gel extraction steps during DNA preparation for microinjections. Better be safe than risk commemorating your sloppiness in a genome of transgenic animal.

Random integration is also frequently accompanied by large-scale structural rearrangements, including deletions, inversions, tandem duplications, and chromosomal translocations. Goodwin et al. found that over 50% of analyzed mouse lines carried chromosomal deletions, while 15 out of 40 also harbored duplications [12]. Similarly, Cain-Hom et al. reported two chromosomal translocations, two cointegrations, and three duplications near the insertion sites in Cre-deleter rodent lines [14]. Numerous other cases involving large tandem duplications have also been described [24,25,26,27]. The underlying reasons for the high frequency of duplications near integration sites remain to be fully elucidated.

Even when such structural changes do not directly affect the phenotype, they can interfere with genotyping, copy number analysis, and transgene detection. Therefore, high-resolution mapping—such as through long-read sequencing (LRS) or TLA—is strongly recommended, even for supposedly “well-characterized” transgenic lines.

**Figure 1 ijms-26-04705-f001:**
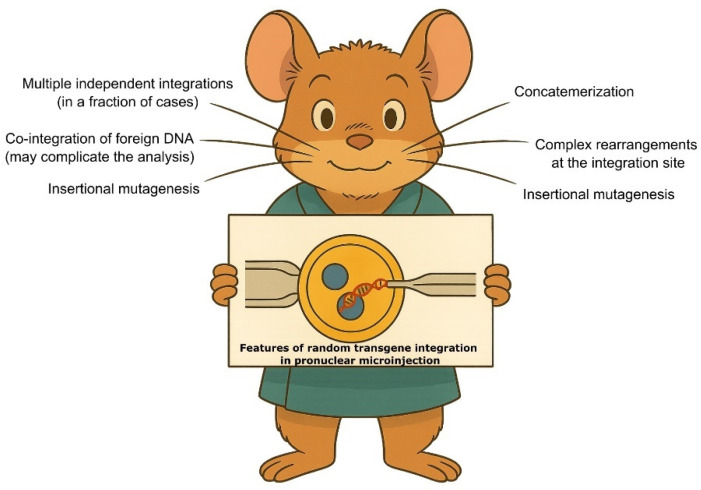
Features of the DNA integration in the pronuclear microinjection to be considered during transgene mapping. These features could complicate the transgene mapping and data analysis (details in the main text). The image was generated with ChatGPT, version GPT4o.

### 1.2. PCR-Based Methods for Transgene Mapping

Classic genome walking approaches include: ligation of universal adapters to linearized genomic DNA (LM-PCR), linear amplification using biotinylated primers (LAM-PCR), circularization of restriction fragments (Inverse PCR (iPCR)), annealing of semi-random primers (e.g., TAIL-PCR, PST-PCR) [1,28]. While these basic principles remain unchanged, many clever modifications have since appeared—ranging from improved degenerate primer designs for TAIL-PCR, to tagmentation-assisted adapter ligation [29], or sonication-based approaches replacing enzymatic restriction in inverse PCR [30]. All of them can be effective for transgene mapping in animals, but without direct meta-analysis under similar conditions, it’s not very useful to discuss each one in detail.

Inverse PCR (iPCR) is one of the earliest and most widely used PCR-based approaches for mapping transgene integration sites [31,32]. Genomic DNA is first digested with restriction enzymes. The resulting fragments, including transgene–genome junctions, are self-ligated to form circularized DNA molecules. Outward-facing primers complementary to the transgene sequence amplify the unknown flanking region. Its efficiency remains remarkably high. For example, in the TRIP-Cas9 project, hundreds of transposon insertions were successfully mapped using iPCR [33]. We have also used iPCR to excise hundreds of transgene copies from a single embryo sample in order to analyze concatemer structures [34].

LM-PCR also involves restriction digestion of genomic DNA, followed by ligation with universal adapters [35]. The transgene-genome junction is amplified via nested PCR using a combination of gene-specific and adapter primers. Unlike iPCR, this method does not require digestion inside the transgene. However, its efficiency is affected by the need for adapter sets tailored to different restriction enzymes. Newer modifications introduce an additional digestion step to eliminate non-specific ligation products, but this requires precise restriction site planning and custom adapter preparation [36]. A recent version uses A-tailing, biotinylated primers, streptavidin capture, and secondary amplification [37]. Splinkerette PCR uses a specially designed hairpin adapter (formed from two ~48/61 nt annealed oligos), which provides greater specificity compared to simple ligation or circularization [38,39]. It has been used in mapping transposon and viral insertion sites [40,41,42] and was recently adapted for mapping integrations in CHO cells with high efficiency [43].

Originally developed for mapping lentiviral integrations, linear amplification mediated PCR (LAM-PCR) uses a biotinylated primer to linearly amplify single-stranded DNA, which is then captured by streptavidin beads [44]. A second strand is synthesized with random primers and digested with restriction enzymes to create a ligation site for PCR adapters. This enrichment strategy improves specificity over standard LM-PCR. Later improvements replaced the restriction step: after capturing the ssDNA, a single-stranded adapter is ligated, and amplification proceeds with two primers [45]. LAM-PCR has also been combined with sonication for deep profiling of viral integration sites [46].

TAIL-PCR remains one of the most accessible transgene mapping tools, especially for beginners. Unlike other methods, it does not require restriction digestion, primer biotinylation, or commercial kits. All that is needed is a few gene-specific primers and a set of arbitrary degenerate (AD) primers, such as 5′-NGTCGASWGANAWGAA-3′. First reaction of TAIL-PCR typically involves the following steps: high-stringency cycles with high annealing temperature to let sequence-specific (SS) primers generate single-stranded DNA, low-stringency cycle (~25 °C) where AD primers bind randomly to genomic DNA, and normal amplification with nested SS and AD primers to enrich transgene–genome junctions. This is followed by nested PCR to improve specificity (Figure 2A).

Originally developed for T-DNA insertion mapping in plants, TAIL-PCR had a 50–70% success rate [47,48]. Later, hiTAIL-PCR improved specificity by optimizing primer structure and PCR cycling [49]. Some reports noted only 20–30% [50] or 39–69% [51] efficiency of this method. Authors demonstrated that pooling classic AD primers in various combinations or designing new AD primers with lower degeneracy levels improved efficiency two-fold. Additional factors that help to improve outcomes include novel processive polymerases, optimizing PCR annealing temperatures, and stronger dilution of the first reaction [52]. Another group observed up to 83% success of TAIL-PCR in mouse transgene mapping even when using original protocol [11]. Compared to alternative methods, TAIL-PCR has a broader range of applications and high efficiency for mapping random insertions in transgenic animals [11,53,54], cell cultures [55], zebrafish [56], and plants [57,58]. Dozens of related methods have emerged based on the same thermal asymmetry principle, including Wristwatch PCR [59], Fork PCR [60], PER-PCR [61], PST-PCR [28]. These modifications aim to reduce non-specific products or extend amplicons beyond 3–4 kb to capture structural rearrangements flanking the integration site.

We recommend classical hiTAIL-PCR using multiple long AD primer pools to minimize the risk of amplifying transgene–transgene junctions (Table 1) [49]. In our own experience, this method worked in over 80% of cases [62,63], later we reanalyzed the uncharted cases with another transgene-specific primers and found end truncations [64]. That said, genomic rearrangements at transgene ends can reduce the reliability of all PCR-based approaches—sometimes, there’s just no primer-binding site at all [65]. Chimeric products due to PCR [66] and ambiguous bands where parts of the transgene map to different chromosomes [67] are not uncommon, so always confirm results with alternative methods like long-distance PCR or LRS.

While PCR-based mapping is not the gold standard anymore in the next-generation sequencing (NGS) era, it still offers valuable solutions for small-scale, cost-sensitive projects. Among them, TAIL-PCR remains our go-to for locating transgene insertions—requiring little more than a few PCRs and Sanger reads.

**Figure 2 ijms-26-04705-f002:**
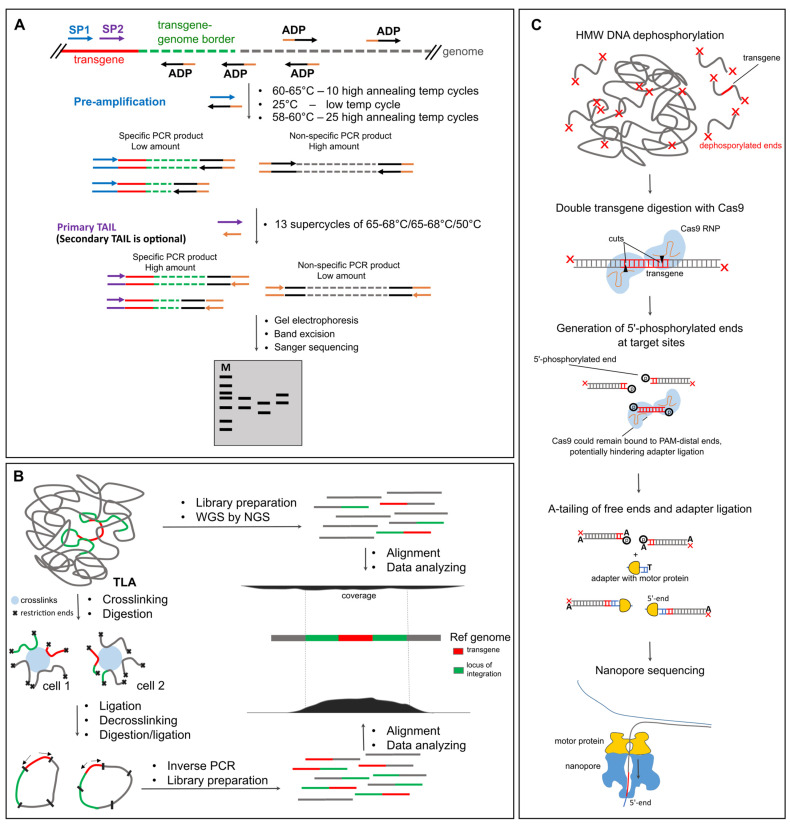
Selected methods for transgene mapping in animals. (**A**) hiTAIL-PCR. The schematic overview of the method shows the main steps which may differ between alternative TAIL-PCR approaches. SP—sequence-specific primer, ADP—arbitrary degenerate primer. (**B**) NGS-based methods: WGS and TLA. (**C**) Nanopore with enrichment by Cas9 digestion.

### 1.3. Next-Generation Sequencing and Target Enrichment

NGS has become an essential part of genomics research [68,69]. Objectively, the most efficient way to identify transgene integration sites is through WGS at sufficient coverage (Figure 2B). But what is the optimal genome coverage for reliable transgene detection? Several studies of transgenic mouse lines have shown that a haploid genome coverage as low as 8× [70] or 11.5× [71] may be sufficient for mapping small insertions. Srivastava et al. reported unsuccessful mapping using standard paired-end sequencing at 18× coverage, and applied mate-pair sequencing instead [72]. WGS is also commonly used for mapping insertions in transgenic farm animals. Zhang et al. sequenced a transgenic cow carrying a human lactoferrin BAC insert. Although the bovine genome was sequenced at ~10× coverage, the effective coverage of the BAC insert reached 20–50× due to concatemerization [54]. However, internal rearrangements made the structure too complex for short-read NGS to resolve. Another study used ultra-deep sequencing (~268×) to analyze a 3.1 kb SRY-GFP construct knock-in [73]. Despite the high coverage, two alleles with complex structural variants had to be resolved using PacBio. Interestingly, the Cas9-linearized vector caused concatemerization but no random integrations outside of the intended “safe harbor” locus [73]. The same group successfully sequenced F1 offspring of a hornless bull with notorious backbone integration [74] at 20× coverage [75]. Transgenic crops are generally sequenced at ~13–14× [76], 21× [77], 29× [78], or even 70× [79], although T-DNA insertions are usually less repetitive and easier to map. These examples illustrate the approximate sequencing depth needed to identify insertion sites. On modern Illumina platforms (e.g., 150 bp paired-end reads), such coverage can still be relatively expensive, especially for large-scale screening (Table 2).

**Table 1 ijms-26-04705-t001:** Cost-effective transgene mapping methods. The efficiency/scalability/cost are given as subjective close estimates based on the literature and personal experience. Cost calculations are presented in Table 2.

	Method 1: hiTAIL-PCR
**Efficiency/Scalability/Cost**	~80%/low/$80 per line
**Advantages**	Simple and cost-effective protocol with high efficiency. Only requires standard PCR, gel electrophoresis and Sanger sequencing. Universal AD primers are compatible with most genomes. Generates relatively long PCR products (300–2000 bp), which improve alignment accuracy over short NGS reads. The hiTAIL-PCR design suppresses non-specific short amplicons [49]. Amplicon length can be extended further with protocol modifications [52].
**Problems**	Requires an intact primer binding site: As with any genome-walking PCR method, successful amplification depends on a functional transgene-specific primer site. If initial attempts fail, new primers spaced every 300–400 bp along the transgene may be required. Non-specific amplification: based on the conditions (transgene copy number, genome complexity, degenerate primer sequence), non-specific amplification may represent a problem. Transgene–transgene junctions are also efficiently amplified and give the misleading characteristic size shift at the gel after the secondary TAIL-PCR. This can be countered by using different AD primers or restriction digestion of the transgene–transgene regions.
**Perfect for**	Single-copy, intact transgene insertions
	**Method 2: WGS by NGS**
**Efficiency/Scalability/Cost**	~95%/average/$250–2400
**Advantages**	Not linked to a specific sequence, making it effective regardless of transgene truncations.
**Problems**	Costly: Achieving 10–30× genome coverage for reliable mapping typically costs over $1000, depending on sequencing provider and genome size. Short read length limits ability to resolve complex integration sites, such as flanking duplications or inversions (which are relatively frequent).
**Perfect for**	Urgent low-scale mapping experiments
	**Method 3: TLA**
**Efficiency/Scalability/Cost**	100%/average/$150–2000
**Advantages**	Uses proximity ligation to enrich for sequences near a known transgene region, increasing the chance of capturing insertion breakpoints with short reads The crosslinking protocol could be established in the lab to enrich NGS data [80], making it one of the most cost-effective mapping methods.
**Problems**	Large constructs (e.g., BACs) or insertions with unknown elements may require multiple primer pairs causing additional expenses. Less accessible than other methods: the protocol involves complex sample preparation and may be more practical through commercial services, which can be expensive and non-transparent.
**Perfect for**	Most cases
	**Method 4: Nanopore LRS + Cas9 enrichment**
**Efficiency/Scalability/Cost**	100%/average/$350–1000
**Advantages**	Long reads enable unambiguous mapping: sequencing reads spanning thousands of base pairs can cover entire integration loci and flanking rearrangements. Cas9-based enrichment could use multiple gRNA increasing coverage efficiency. Protocols for large scale in vitro gRNA synthesis from PCR templates are simple and fast [81].
**Problems**	Requires high-molecular-weight (HMW) DNA: Extraction protocols are technically demanding and require fresh or well-preserved samples. Degraded short-length DNA or overly viscous samples can ruin flow cell performance. Even with enrichment, coverage may be limited. Nanopore error rates (~1%) can be problematic for distinguishing barcoded or repetitive sequences [82]
**Perfect for**	Multicopy concatemers, complex insert sites

To improve detection sensitivity and reduce sequencing costs, target enrichment techniques have been developed to increase the proportion of reads covering transgene-genome junctions. Since an insertion site represents only a tiny fraction of the mouse genome, sequencing a 10 kb transgene at high coverage (>10×) requires just a few thousand reads—an insignificant portion of a typical NGS dataset (Table 2). Many enrichment methods have emerged (see recent review [1]), some based on earlier molecular biology strategies such as LAM-PCR [83], TAIL-PCR [84], inverse PCR [85], while others involve newer approaches like chromatin crosslinking or Cas9-mediated enrichment [86]. All these are used to enrich sequencing libraries prior to high-throughput sequencing [87]. In this section, we briefly describe several popular enrichment methods applicable to transgene mapping: biotinylated probe capture (hybrid capture), chromatin-crosslinking (TLA), and others. The final choice depends on user expertise and available resources.

Perhaps the most widely used enrichment technique for mapping transgene integration is TLA [80]. TLA builds on the principles of chromatin conformation capture (3C/Hi-C). In the first step, formaldehyde crosslinks chromatin, fixing together DNA regions that are physically close—including the transgene and flanking genomic sequences. Next, the DNA is digested with a frequent-cutting enzyme (e.g., NlaIII), followed by religation under dilute conditions to promote intramolecular ligation. After reverse crosslinking, a second round of digestion and religation produces circular DNA molecules enriched in ligation products near the transgene. PCR with outward-facing transgene-specific primers amplifies these circles, allowing selective enrichment of flanking genomic regions. The resulting fragments are subjected to standard library preparation and sequencing (Figure 2B). Although the TLA protocol appears complex, it can be performed in any lab with modest resources [80,88,89,90]. However, data analysis requires proficiency in interpreting chromatin ligation-based datasets. For this reason, many researchers outsource TLA mapping to commercial providers like Cergentis [13,91,92,93,94]. TLA has proven particularly valuable in large-scale transgenic mouse studies [12].

Typically, the region with the highest coverage—often exceeding 100 kb—indicates the most likely insertion site. Large constructs like BACs may require 5–6 primer sets and rounds of TLA to achieve sufficient coverage [13,93]. A major advantage of TLA is that the resulting amplicons contain not only flanking sequences but also the entire transgene. Also, because homologous chromosomes occupy distinct nuclear territories, TLA is also capable of haplotyping, detecting SNVs, and identifying large structural variants near integration sites. However, the use of short Illumina reads (~150 bp) limits resolution in repetitive regions and fails to fully resolve complex concatemers. Combining TLA with LRS can improve structural resolution: transgene flanks identified by TLA can guide Cas9 digestion and Nanopore-based enrichment [95,96].

Another widely used method is hybrid target capture [1], including solid-state microarrays [97] and magnetic beads. The latter approach is more convenient. Biotinylated DNA or RNA probes anneal to denatured, fragmented genomic DNA. Hybridized molecules are captured using streptavidin-coated magnetic beads. The captured DNA is then extended by polymerase to complete sequencing templates. A major advantage of hybrid capture is that overlapping 60–120 nt probes can cover an entire transgene sequence—especially useful for random integration mapping, because the borders of the insert could be truncated. This method has been used successfully in multiple studies [22,98], and is considered cost-effective once probes are synthesized (Table 2). For instance, Magembe et al. used a pool of 413 xGen Lockdown probes to tile an 18 kb T-DNA region in plants. They found around 10–20% of target reads in the NGS data. Although probe coverage was uneven, 30 and 27 of each of the T-DNA ends from 34 lines were successfully mapped [98]. In another study, capture probes targeted bovine leukemia virus (BLV) insertions with modest enrichment: 10.2% of the total reads mapped to the target proviral genome [99]. Iwase et al. used hybrid enrichment to detect HIV-1 integration sites and generated around 5% of the target reads of the total data [100].

An intriguing and recent addition to the toolbox is T7-based transcriptional mapping, used by Li et al. for locating transposon insertions [101]. This method requires addition of a ~20 bp T7 promoter near the end of the transgene. Genomic DNA is subjected to in vitro transcription, followed by cDNA synthesis using random primers—eliminating the need for restriction digestion or ligation. Although the effective read length depends on the transcription reaction, cDNAs can exceed 1 kb, enabling efficient transgene-genome junction recovery. This approach is promising but may suffer from loss of the T7 sequence during random integration events.

In summary, NGS is a powerful tool for mapping transgene insertions. For many applications, WGS or commercial TLA remains the best choice (Figure 2B), depending on budget and available expertise (Table 1). However, the limited read length of short-read platforms often complicates mapping—especially for rearranged or repetitive regions. For example, in the study by Siddique et al., only one end of a T-DNA insertion was resolved even at 36× coverage [102]. In another report, Peng et al. mapped a complex insertion in a repetitive region of the maize genome but even 41× WGS and TAIL-PCR failed to identify the region, which required long-read sequencing [103]. For transgenic core facilities or large-scale mouse projects, implementing enrichment protocols such as hybrid capture or TLA can significantly improve mapping outcomes. In this review, we only scratched the surface of available tools. While many protocols are low-cost, they require significant optimization and bench skills. Still, for researchers who can manage custom biotinylated probe synthesis or chromatin crosslinking, the results are often worth the effort.

As one colleague once remarked, during yet another transgene mapping crisis: “What am I supposed to do with these 100 bp snippets? Give me long reads or I’m out!”.

**Table 2 ijms-26-04705-t002:** Comparison of costs and labor time for transgene mapping methods. Estimates are based on a hypothetical 10 kb transgene and should be adjusted according to the expected insert size. Pricing and time estimates exclude DNA isolation and do not account for delivery time, which may vary significantly depending on geographic location. High-throughput sequencing using platforms such as Revio (PacBio), PromethION (Nanopore), and NovaSeq 6000 (Illumina) is typically outsourced to specialized service providers rather than conducted in individual laboratories. Therefore, when planning such experiments, it is essential to consider additional factors, including probe design and synthesis time, shipping logistics, and service turnaround—each of which can substantially affect the overall project cost and timeline.

	Preparation Price per Sample *	Sequencing Price per Sample	Sufficient Sequence Data (Gb)/On-Target Data (%)/On-Target Coverage (Reads)	Preparation/Run Time
**Inverse PCR** [32]	$20–$30	$10–$30 (Sanger)	<0.001 Gb/NA/NA	~9–12/3–4 h
**TAIL-PCR** [49]	$40–$50	$10–$30 (Sanger)	<0.001 Gb/NA/NA	~8–12/3–4 h
**WGS by NGS** (Illumina paired-end 150 bp) [54,70,73,75]	$75–$135	NGS Option A: NovaSeq 6000 S4 ~ $160–$250 NGS Option B: NextSeq 500/550 ~$1900–$2400	30 Gb/<0.01%/>10	NGS Option A: ~3–5/45 h NGS Option B: ~3–5/35 h
**NGS + TLA** (commercial) [12,13,93]	$1000–$2000		NA	Weeks
**NGS + TLA** (lab) [88,89,90]	$50–$75	NGS Option A: ~ $35–$70 NGS Option B: ~ $200–$250	3 Gb/~30–70%/>30	NGS Option A: ~36–48/35 h NGS Option B: ~36–48/45 h
**NGS + hybrid capture** (using 120 nt commercial tiling probes) [73,74]	$180–$250	NGS Option A: ~ $10–$20 NGS Option B: ~ $75–$150	1 Gb/~40–80%, up to 95% **/>30	NGS Option A: ~24–36/45 h NGS Option B: ~24–36/35 h
**NGS + hybrid capture** (probes made in the lab)	$50–$60	NGS Option A: ~ $10–$20 NGS Option B: ~ $75–$150	1 Gb/~80–90%, up to 93% **/>50	NGS Option A: ~50/45 h NGS Option B: ~50/35 h
**NGS + T7 In vitro transcription** [101]	$50–$70	NGS Option A: ~ $35–$70 NGS Option B: ~ $200–$250	3 Gb/~35–70%/>30	NGS Option A: ~6–9/45 h NGS Option B: ~6–9/35 h
**PacBio WGS** [34,104]	~ $100–$150	$900–$1600	45–90 Gb/>0.01%/>15–25	6–10/24–36 h
**PacBio + hybrid capture** (using 120nt commercial probes) [105]	~ $350–$500	$125–$200	5–10 Gb/40–60%/>30	30–40/24–36 h
**Oxford Nanopore Technologies (ONT) WGS** [15,106,107]	~ $100–$150	ONT Option A: MinION, 2–3 flow cells $1200–$2400 ONT Option B: PromethION (shared) $300–$600	60–90 Gb/>0.01%/20–30	ONT Option A: ~5–7/24–60 h ONT Option B: ~5–7/48–72 h
**ONT + nCATs** [26,96,108]	~ $160–$200	ONT Option A (1 flow cell): $600–$800 ONT Option B: $100–$150	30 Gb/10–40% ***/20–30	ONT Option A: ~7–10/24–60 h ONT Option B: ~7–10/48–72 h
**ONT + internal cuts** (AFIS-seq, CRISPR-LRS) [27,46,109]	$150–$200	ONT Option A (1 flow cell): $600–$800 ONT Option B: $100–$150	30 Gb/5–40% ***/>30	ONT Option A: ~7–10/24–60 h ONT Option B: ~7–10/48–72 h
**Nanopore** + **Xdrop** (commercial) [16,110,111]	$650–$900	ONT Option A (1 flow cell): $600–$800 ONT Option B: $100–$150	10 Gb/~60–90%/>30	ONT Option A: ~4–5 days/24–60 h ONT Option B: ~4–5 days/48–72 h

* The price includes NGS library preparation, along with quality and quantity control. For Sanger-based methods, the price includes enzymatic reactions and dideoxynucleotide triphosphates labeled with fluorescent dyes. ** Depends on multiple parameters related to probe quality. *** Depends on gRNA efficiency.

### 1.4. Long-Read Sequencing

In recent years, two independent platforms—PacBio (Pacific Biosciences) and Oxford Nanopore Technologies—have developed third-generation sequencing (TGS), also referred to as single-molecule sequencing (SMS) or LRS [82,112]. These technologies routinely produce reads in the 10–100 kb range and avoid PCR-associated artifacts. LRS has been successfully applied for genome polishing [107], sequencing of repetitive chromosome regions [113], and even for whole-genome assembly from single sandflies [114]. Novel applications include RNA isoform sequencing and epigenetic modifications measurements, combined with single-cell sequencing approaches [112,115,116]. Here, we focus on the use of LRS for transgene mapping and concatemer structure analysis.

The PacBio platform is based on single-molecule real-time (SMRT) sequencing. Fragmented DNA is ligated to single-stranded hairpin adapters from both sides, and a sequencing primer anneals to the hairpin region. Fluorescently labeled nucleotides allow base detection as docked polymerase molecule replicates the circularized DNA in a special well (SMRT cell). PacBio reads are typically limited to 25–30 kb, so that the circular consensus sequencing (CCS) strategy enables multiple polymerase passes over the same molecule, greatly increasing accuracy [117]. PacBio has been used for transgene mapping in mice [34,118] and plants [105], although it is less frequently chosen than Nanopore. When comparing the two platforms, the CCS mode of PacBio offers superior fidelity (>99%) compared to earlier generations of Nanopore sequencing (~90–95%) [112,117]. Moreover, Nanopore sequencing is particularly prone to errors in homopolymer regions [119]. However, the error rate is not of primal importance for transgene mapping, because long read length compensates for errors. The cost of both LRS platforms continues to fall and is now broadly comparable [82], depending on the specific instrument (Table 2). A comprehensive and critical comparison of the two LRS methods is provided in a recent review of Schell et al. [117].

Oxford Nanopore sequencing works by measuring ionic current changes as DNA moves through a biological nanopore embedded in a membrane [120]. This enables extremely long (megabase) reads, although average read lengths are typically similar to PacBio. Different authors casually report long reads around 200 kb [27], 238 kb [106], or 351 kb [121]. Occasionally such reads could contain transgenes and provide valuable insight into concatemer structure.

Below are selected examples to guide Nanopore-based experimental planning. Technology and chemistry improvements are ongoing, but for most transgene mapping experiments, a single MinION flow cell (typically R9 series) can produce 5–10 Gb of data—sufficient for a typical animal or plant transgenic line. In one early study, Nicholls et al. generated 4.88 Gb (1.8× haploid genome coverage) using a MinION run that yielded 611,279 reads with an N50 of 28 kb [15]. Among these, 25 reads contained transgene fragments, but only one 5.5 kb read spanned the genome–transgene junction within a 450 kb concatemer [15]. Suzuki et al. used a single MinION flow cell to sequence a transgenic mouse, obtaining 3 Gb of data (1× hgc; 922,210 reads; N50 = 7.6 kb). A 21.5 kb read covering one and a half copies of the transgene allowed successful integration mapping [106]. Another group investigated Cre-deleter mouse lines that failed to yield homozygotes in PCR screenings. TLA identified a 95 kb tandem duplication close to the floxed cassette in the gene of interest with unedited coding sequence. Three Nanopore runs produced 13 Gb (4.4× hgc; 699,343 reads; N50 = 40.7 kb), identifying 9 on-target reads and unambiguously resolving the rearrangement [25]. Giraldo et al. sequenced transgenic crops using one flow cell per sample and obtained 7.3–10.4 Gb with sufficient on-target coverage, though average read lengths varied from 1.6 to 12 kb [122]. In a soybean study, Li et al. generated 2.8 Gb (2.5× hgc; 1,061,117 reads) and found two reads spanning transgene–genome junctions. The results confirmed the site previously mapped by TAIL-PCR, highlighting the latter’s cost-efficiency [121].

These examples illustrate that running a single MinION may yield only a few useful reads and become a costly endeavor as transgenes represent only ~0.01% of the genome. Enrichment strategies are often necessary when working with transgene mapping. In contrast to NGS-based enrichment methods, LRS approaches must preserve long DNA fragments. Two commonly used strategies—hybrid capture and Cas9 digestion—are compatible with LRS [82,123].

For PacBio, DNA is usually fragmented and size-selected to ~10–20 kb, while Nanopore sequencing often uses high-molecular-weight DNA [124]. Biotinylated probe enrichment for PacBio has been used to enrich symbiont genomes by 11–200× [125] or blood group system loci by 737× [124]. Biotin-based PacBio enrichment method, LIFE-seq, was introduced by Zhang et al. [105]. This method uses 75 nt tiling probes to cover known plasmid sequences (~99% coverage). Seven transgenic crop samples were enriched and sequenced, yielding 1.8–2.7 Gb per sample. On average, 17,000–25,000 unique CCS reads (average length ~6 kb, N50 ~17 kb) were obtained [105]. These data enabled mapping of insertion sites and partial concatemer reconstruction. Biotin enrichment was also applied to Nanopore sequencing. In the soybean study mentioned earlier, enrichment allowed identification of 51 transposon integration sites from a single Nanopore flow cell [121]. Although probe synthesis is costly and may reduce read length during sample preparation [125], this strategy avoids transgene fragmentation and does not require preservation of transgene ends. Other enrichment strategies for LRS include sonication-based inverse PCR (SIP) [30] and TLA-seq [126], although these are complex and less standardized than Cas9-based methods.

The CRISPR/Cas9 system has become a favored tool for target enrichment. In this approach, guide RNAs define cleavage points in the genome or transgene, producing ligation-compatible ends. Though PacBio-compatible [127,128], most applications in transgene mapping use Nanopore. One widely adopted method is nCATS (Nanopore Cas9-Targeted Sequencing), where high-molecular-weight DNA is dephosphorylated, treated with Cas9–gRNA RNPs, and only the phosphorylated cut ends are ligated to Nanopore adapters [108]. nCATS method has become very popular for human diagnostics with enrichment of targeted regions of 25× [129], 665× [130], >100× [131]. Enrichment is especially useful for LRS in clinical samples with heterogenous cell populations or low target DNA quantity [119,132].

nCATS has been successfully applied to transgene mapping in various organisms [26,95,133]. Low et al. used nCATS to confirm site-specific integration of human ACE2 transgene into the Rosa26 locus via Bxb1-mediated recombination. With one flow cell they achieved 195× coverage of an 8.5 kb cassette [26]. In the same study, they sequenced mouse line with random multicopy integration of a similar transgene, and two 70–80 kb contigs were identified which contained the transgene-genome borders [26]. Other group compared nCATS, TLA, and Southern blotting to map transgene insertions in CHO cells [96]. For small transgenes (3–6 copies), nCATS produced contigs up to 41.6 kb from 22 reads and successfully resolved rearrangements. Notably, this allowed confirmation of peculiar Southern blot results obtained earlier—demonstrating the continuity of two mapping technologies [96]. nCATS is now supported by an official Nanopore protocol, but it requires prior knowledge of flanking sequences and would not be useful for initial transgene mapping.

Alternative Cas9 enrichment method is based on the same principle but DNA is digested inside the transgene region (Figure 2C). Funnily enough, this otherwise straightforward approach still lacks a definitive and concise acronym. The method is inconsistently named across publications and is referred to as “Targeted Cas9 sequencing” in the official Nanopore protocol—a term easily confused with nCATS, which, unlike this method, requires prior knowledge of the flanking sequences. For clarity, we propose a temporary name: CHAD (CRISPR-based Homing for Anchored Detection). Given how much scientists enjoy inventing acronyms—see the many creative efforts for TAIL-PCR modifications—it might be time to standardize the terminology, especially considering the growing popularity of the CHAD approach. One of the first applications of this strategy was AFIS-seq (Amplification-Free Integration Site sequencing), which mapped lentiviral integrations using paired Cas9 cuts inside the transgene. Enrichment ranged from 285–1612×, with average read lengths of ~12 kb [46]. In comparison to NGS-based S-EPTS/LM-PCR method, AFIS-seq provided fewer ambiguous reads thanks to longer sequencing length. McDonald et al. applied CHAD with a single cut to human samples to study mobile elements. One flow cell yielded ~110,000 reads, 31% of which were on-target (54× enrichment; N50 = 25 kb) [134]. Similarly, Hertel et al. used dual cuts flanking eGFP transgene in CHO cells, achieving 86–244× enrichment and revealing unplanned random integrations [135]. Bryant et al. used CRISPR-LRS with paired gRNAs to map several transgenes in mice. For a 217 kb BAC, 9 reads (0.03%) spanned transgene-genome borders [27]. With extra guides, enrichment improved to 0.15–0.35%. However, internal concatemer structure was lost due to Cas9 fragmentation: in the Sm22-Cre mouse line where qPCR detected ~20 copies, Nanopore only detected a max of 4 per read [27]. Ironically, WGS of the ultra-high molecular weight (HMW) DNA with Nanopore generated more useful detail in a few reads (6 selected reads, 89 kb average read length) than Cas9 enrichment due to the longer read sizes [27]. We also applied CHAD to a 5 kb hACE2 concatemer (~70 copies). Nanopore WGS (0.25× genome coverage) yielded 15 transgene reads, while CHAD produced 864 reads longer than 3 kb, mapping one border at the cost of losing internal concatemer structure [109]. We suspect that reads with the second transgene-genome border were lost because we enriched with only one Cas9 site instead of two (Figure 2C). Importantly, Cas9 often blocks the protospacer adjacent motif (PAM)-distal end [136], hindering adapter ligation and reducing coverage in the respective direction by 2–10× [129,130,137]. Thermolabile Proteinase K treatment [137] or using Cpf1, which does not block ends, may help to improve nuclease-based targeting [138].

Finally, a novel Nanopore-compatible method, Xdrop, offers an original approach to target enrichment [110,139]. In this technique, the target locus is captured indirectly using a short PCR amplicon that is designed to lie within or near the region of interest, such as a transgene. HMW genomic DNA is mixed with PCR reagents and primers, and encapsulated in droplets using an oil emulsion system. During the droplet PCR fluorescence is triggered by an intercalating dye only in droplets that contain the specific target DNA. Typically, only about 0.01% of the double emulsion droplets will contain the desired fragment. These fluorescent droplets are then isolated via fluorescence-activated cell sorting (FACS) () and subjected to single-molecule multiple displacement amplification (dMDA) to amplify the enriched genomic DNA. The resulting product is then sequenced using the Nanopore platform [110,139].

Early publications have already demonstrated the successful use of this method to map transgenes in mice [110] and plants [111], as well as to detect complex genomic rearrangements in human cells [16,110]. These studies show that indirect targeting by droplet PCR provides very high enrichment levels (100× to 3000×) and enables detailed resolution of internal rearrangements, albeit at the cost of reduced average read length (around 5 kb) [110]. Given the technical complexity and specialized instrumentation involved, it is unlikely that Xdrop will be used routinely for mapping transgenes in animal models. However, one clear advantage is that indirect enrichment preserves the internal structure of concatemers, which is often lost in Cas9-based methods.

Ultimately, we would recommend the CHAD approach for most transgene mapping scenarios (Table 1). While a typical Nanopore run on a standard flow cell may yield only 3–5 reads per million reads covering a transgene border—sometimes with no guarantee of successful mapping—Cas9 enrichment offers a more targeted and controlled strategy, and it is not especially difficult to implement. One full run using this method requires a single flow cell (~$800) and a library prep kit (~$200), both of which can potentially be reused, making it cost-effective for many labs (Table 2). Unfortunately, CHAD destroys the internal concatemer structure unlike the original nCATS, where the cuts are introduced in the flanking sequences and preserve the concatemer structure, with up to 30–100 kp inside concatemer, which could be enough to assemble the whole insert, depending on the transgene size [96].

Compared to PCR-based techniques, there are few disadvantages to LRS, aside from the requirement for larger quantities of high-molecular-weight genomic DNA (in the microgram range), which is usually not a problem when working with animal tissue, but may be a problem with valuably founders or tiny model animals. At the same time, it’s important to note that current Cas9 enrichment workflows generally lead to low sequencing coverage, making them unsuitable for applications requiring single-nucleotide resolution, such as precise indel detection or barcode identification.

## 2. Conclusions

Transgene mapping remains a critical yet technically diverse task, with no universal solution. In this review, we evaluated a range of available methods—from classic PCR-based genome walking to advanced enrichment protocols for NGS and LRS (Figure 3). For small-scale projects or initial screening, we recommend hiTAIL-PCR as a low-cost and accessible method. It requires minimal optimization and demonstrates high success rates, especially when transgene ends are preserved. For reliable integration site search, WGS and TLA allow high-throughput mapping, though they could be costly and typically require access to sequencing facilities and bioinformatics support. When long-range information is essential, particularly in concatemer inserts or rearranged regions, Nanopore sequencing combined with Cas9-based enrichment (e.g., CHAD) is currently the most promising approach, because it enables sequencing of long DNA fragments for easy alignments. However, LRS methods are still evolving and can be technically demanding, with variable enrichment efficiency and sensitivity to DNA quality. Looking ahead, the future of transgene mapping is promising. Perhaps in five years emerging techniques such as adaptive sampling [82], AI-enhanced base calling [140], and real-time alignment filtering [141] will likely make LRS accessible and targeted to specific regions. This will signify the end of the old genome walking era, but until then we have to keep walking.

## Figures and Tables

**Figure 3 ijms-26-04705-f003:**
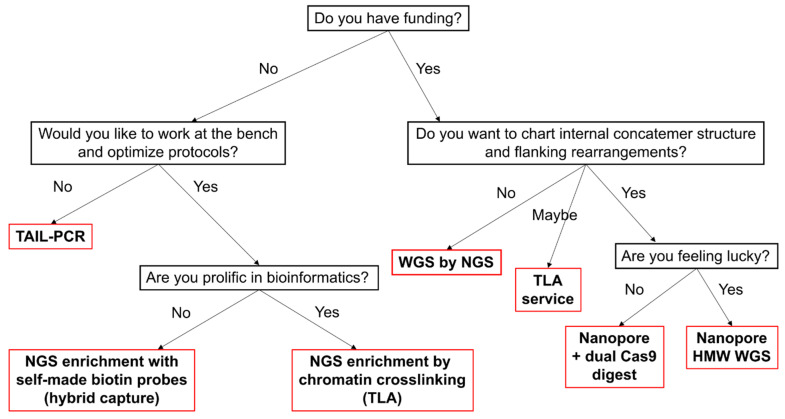
Decision tree for choosing a mapping method.

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
