# Peer review of "Transgene Mapping in Animals: What to Choose?"

_ijms, 2025, doi:10.3390/ijms26104705_

Round 1
Reviewer 1 Report
Comments and Suggestions for Authors
This review provides a systematic summary of transgene mapping methods in animals, covering classic PCR-based techniques, next-generation sequencing (NGS), and long-read sequencing (LRS) platforms. The structure is clear, the content is thorough, and the article serves as a valuable reference for researchers selecting appropriate methodologies. The logical flow and accessible language, supported by tables and figures, enhance readability. However, certain details require refinement to improve rigor and comprehensiveness.
1. Some cited studies are outdated (e.g., 2011 review). Update with recent advancements (e.g., 2023 Cas9 enrichment optimizations, single-cell sequencing applications).
2. Clarify NGS cost estimates (e.g., commercial vs. in-lab expenses) and address regional/platform variations.
3. The authors recommend hiTAIL-PCR, WGS, TLA, and Nanopore+CRISPR as "top four methods," but limitations (e.g., hiTAIL-PCR’s reliance on intact primer sites, Nanopore’s DNA quality requirements) are underdiscussed.
4. Include specific use-case examples (e.g., LRS for multicopy concateners vs. TAIL-PCR for single-copy insertions).
5. TAIL-PCR success rates vary widely (20%–83%) across studies. Elaborate on factors influencing outcomes (e.g., genome complexity, primer design).
6. Figure 2C ("Nanopore with Cas9 digestion") requires clearer labeling of key steps (e.g., Cas9 cut sites, adapter orientation).
7. Standardize units in Table 1’s "Total / On-target data (coverage)" column (e.g., "GB" or "%") and clarify whether "coverage" refers to genome-wide or targeted regions.
8. Simplify the decision tree (Figure 3) by merging overlapping criteria (e.g., "bioinformatics proficiency" and "outsourcing options").
9. Redundancies exist (e.g., historical context of PCR methods in Introduction and Section 1.3). Streamline for conciseness.
Author Response
Reviewer 1:
- Some cited studies are outdated (e.g., 2011 review). Update with recent advancements (e.g., 2023 Cas9 enrichment optimizations, single-cell sequencing applications).
We replaced outdated references (e.g., Hui et al., 2002; Tonooka & Fujishima, 2009; Van Dijk et al., 2018) with more recent reviews (2020–2025), including Hook & Timp (2023), Schell et al. (2025), Singh (2022), and Brlek et al. (2024). However, we retained one mention of Leoni (2011) as a historical reference to illustrate the boom of genome walking methods; the text now acknowledges its outdated.
We also expanded the LRS section to include emerging applications such as:
- RNA isoform sequencing and epigenetic modification detection
- Integration with single-cell sequencing (Warburton & Sebra, 2023; Gordon et al., 2024; Liu & Conesa, 2025)
We also added some remarks to the LRS section:
“Novel applications include RNA isoform sequencing and epigenetic modifications measurements, combined with single-cell sequencing approaches (Warburton, Sebra, 2023; Gordon et al., 2024; Liu, Conesa, 2025)”.
“When comparing the two platforms, the CCS mode of PacBio offers superior fidelity (>99%) compared to earlier generations of Nanopore sequencing (~90–95%) (Warburton & Sebra, 2023; Schell et al., 2025). Moreover, Nanopore sequencing is particularly prone to errors in homopolymer regions (Wongsurawat et al., 2020). However, the error rate is not of primal importance for transgene mapping, because long read length compensates for errors. The cost of both LRS platforms continues to fall and is now broadly comparable (Hook. Timp, 2023), depending on the specific instrument (Table 2). A comprehensive and critical com-parison of the two LRS methods is provided in a recent review of Schell et al (Schell et al,. 2025)”.
“Enrichment is especially useful for LRS in clinical samples with heterogenous cell populations or low target DNA quantity (Wongsurawat et al., 2020; Cottingham et al., 2025)”.
- Clarify NGS cost estimates (e.g., commercial vs. in-lab expenses) and address regional/platform variations.
We have added additional clarifications for Table 2 (former Table 1):
* The price includes NGS library preparation, along with quality and quantity control. For Sanger-based methods, the price includes enzymatic reactions and dideoxynucleotide triphosphates labeled with fluorescent dyes.
** Depends on multiple parameters related to probe quality
*** Depends on gRNA efficiency
Regional variations are pretty hard to estimate, which is mentioned in Table headline («High-throughput sequencing using platforms such as Revio (PacBio), PromethION (Nanopore), and NovaSeq 6000 (Illumina) is typically outsourced to specialized service providers rather than conducted in individual laboratories. Therefore, when planning such experiments, it is essential to consider additional factors, including probe design and synthesis time, shipping logistics, and service turnaround—each of which can substantially affect the overall project cost and timeline»).
The authors recommend hiTAIL-PCR, WGS, TLA, and Nanopore+CRISPR as "top four methods," but limitations (e.g., hiTAIL-PCR’s reliance on intact primer sites, Nanopore’s DNA quality requirements) are underdiscussed.
We have expanded the description of four top methods from Fig.2 as a table (new Table 1). We have tried to better expand the pros and cons of each method. We have also added the use-case examples («optimal for»).
Include specific use-case examples (e.g., LRS for multicopy concateners vs. TAIL-PCR for single-copy insertions).
See previous response.
TAIL-PCR success rates vary widely (20%–83%) across studies. Elaborate on factors influencing outcomes (e.g., genome complexity, primer design).
Indeed, some articles reported low efficiencies (which may or may not be connected with the fact that the researches wanted to highlight their improvements). As you suggested, we mentioned the factors influencing the variable success rates of TAIL-PCR:
“Some reports noted only 20–30% [48] or 39–69% [49] efficiency for this method. The authors demonstrated that pooling classic AD primers in various combinations, or designing new AD primers with lower degeneracy levels, improved efficiency twofold. Additional factors that help improve outcomes include the use of novel processive polymerases, optimization of PCR annealing temperatures, and stronger dilution of the first reaction (Jia et al., 2017)”.
We added some of the factors to the Table 1 for better visibility.
Figure 2C ("Nanopore with Cas9 digestion") requires clearer labeling of key steps (e.g., Cas9 cut sites, adapter orientation).
We have reworked Fig.2C and added more details. The text was moved to new Table 1.
Standardize units in Table 1’s "Total / On-target data (coverage)" column (e.g., "GB" or "%") and clarify whether "coverage" refers to genome-wide or targeted regions.
Fixed.
- Simplify the decision tree (Figure 3) by merging overlapping criteria (e.g., "bioinformatics proficiency" and "outsourcing options").
Thanks for the suggestion, however, in all honesty, we could not understand the thought. Is the main suggestion to introduce the first bifurcation at bioinformatics proficiency vs outsourcing options? We think that rooting at funding is more realistic and ironic. Also, simplifying the tree could make it more boring with less options. So, respectfully, we’d leave the picture as is, unless Reviewer in insisting and willing to explain the task in more detail.
- Redundancies exist (e.g., historical context of PCR methods in Introduction and Section 1.3). Streamline for conciseness.
We have moved redundant content about PCR history from 1.3 to introduction and merged it with other text.

Reviewer 2 Report
Comments and Suggestions for Authors
This manuscript addresses an important topic regarding the selection of methods for mapping transgene DNA integration sites; however, it needs to be strengthened in the following points.
-Organization and Focus: The review provides extensive technical details but lacks a systematic, criteria-based comparison of methods. A clearer justification for prioritizing specific techniques (e.g., TAIL-PCR, TLA, WGS, Nanopore) is needed.
-Recommendation Criteria: Method selection should be assessed based on defined performance parameters such as efficiency, resolution, and scalability, rather than anecdotal observations.
-Referencing: Practical claims (e.g., success rates, enrichment efficiencies) should be supported with appropriate references throughout the text.
-Critical Discussion: The advantages and limitations of long-read sequencing technologies should be discussed more critically, rather than descriptively. And should include more AI-assisted mapping site selection literature for discussion.
-Language and Tone: The manuscript occasionally uses conversational expressions that detract from scientific rigor (such as “Note to self”, “sigh!” ). A more neutral and professional tone is recommended.
Comments on the Quality of English LanguageNeeds to focus on the grammar and expression.
Author Response
Reviewer 2:
This manuscript addresses an important topic regarding the selection of methods for mapping transgene DNA integration sites; however, it needs to be strengthened in the following points.
-Organization and Focus: The review provides extensive technical details but lacks a systematic, criteria-based comparison of methods. A clearer justification for prioritizing specific techniques (e.g., TAIL-PCR, TLA, WGS, Nanopore) is needed.
Thank you for this valuable suggestion! We acknowledge that, as noted in the manuscript, meta-analyses for genome walking methods are lacking, and not all methods reviewed have been used for animal transgene mapping. We decided to create additional table (New Table 1 in the text) to better describe the selected methods, including recommendation from your second point.
-Recommendation Criteria: Method selection should be assessed based on defined performance parameters such as efficiency, resolution, and scalability, rather than anecdotal observations.
See previous comment.
-Referencing: Practical claims (e.g., success rates, enrichment efficiencies) should be supported with appropriate references throughout the text.
We have carefully revised the manuscript to support all practical claims with appropriate citations. For example, we added sources supporting estimates such as “at frequencies of 0.1–1% per DSB.” Where numerical values remain subjective, we now clearly indicate this in the text to maintain transparency.
-Critical Discussion: The advantages and limitations of long-read sequencing technologies should be discussed more critically, rather than descriptively. And should include more AI-assisted mapping site selection literature for discussion.
Thanks, we have added more discussion about the LRS methods.
We have revised the LRS section to include a more critical comparison of technologies and incorporated new references. The advantages and drawbacks of LRS are now also reflected in Table 1. We updated the narrative with summary sentences and added relevant recent literature, such as Schell et al. (2025) and Hook & Timp (2023), which critically analyze platform fidelity, pricing, enrichment methods, and availability. Moreover, we incorporated references to AI-assisted sequencing algorithms in the conclusion section.
We also added some remarks to the LRS section:
“Novel applications include RNA isoform sequencing and epigenetic modifications measurements, combined with single-cell sequencing approaches (Warburton, Sebra, 2023; Gordon et al., 2024; Liu, Conesa, 2025)”.
“When comparing the two platforms, the CCS mode of PacBio offers superior fidelity (>99%) compared to earlier generations of Nanopore sequencing (~90–95%) (Warburton & Sebra, 2023; Schell et al., 2025). Moreover, Nanopore sequencing is particularly prone to errors in homopolymer regions (Wongsurawat et al., 2020). However, the error rate is not of primal importance for transgene mapping, because long read length compensates for errors. The cost of both LRS platforms continues to fall and is now broadly comparable (Hook. Timp, 2023), depending on the specific instrument (Table 2). A comprehensive and critical com-parison of the two LRS methods is provided in a recent review of Schell et al (Schell et al,. 2025)”.
“Enrichment is especially useful for LRS in clinical samples with heterogenous cell populations or low target DNA quantity (Wongsurawat et al., 2020; Cottingham et al., 2025)”.
-Language and Tone: The manuscript occasionally uses conversational expressions that detract from scientific rigor (such as “Note to self”, “sigh!” ). A more neutral and professional tone is recommended.
We appreciate this feedback. While we initially used conversational phrases to create a more accessible tone, we recognize the need for scientific rigor and have removed expressions such as “Note to self,” “sigh!”, “Let’s be honest,” “to say the least,” and similar phrases.

Round 2
Reviewer 1 Report
Comments and Suggestions for Authors
Thank you for submitting the revised version of your manuscript. I appreciate the thorough revisions and detailed responses to the review comments.
Reviewer 2 Report
Comments and Suggestions for Authors
The authors have addressed all the points, congrats.